# Exploiting MMD and Sinkhorn Divergences for Fair and Transferable Representation Learning

**Luca Oneto**
University of Genoa
luca.oneto@unige.it

**Michele Donini**
Amazon Web Services
donini@amazon.com

**Giulia Luise**
University College London
g.luise.16@ucl.ac.uk

**Carlo Ciliberto**
Imperial College London
c.ciliberto@imperial.ac.uk

**Andreas Maurer**
Isitituto Italiano di Tecnologia
am@andreas-maurer.eu

**Massimiliano Pontil**
Isitituto Italiano di Tecnologia & University College London
massimiliano.pontil@iit.it

## Abstract

Developing learning methods which do not discriminate subgroups in the population is a central goal of algorithmic fairness. One way to reach this goal is by modifying the data representation in order to meet certain fairness constraints. In this work we measure fairness according to demographic parity. This requires the probability of the possible model decisions to be independent of the sensitive information. We argue that the goal of imposing demographic parity can be substantially facilitated within a multitask learning setting. We present a method for learning a shared fair representation across multiple tasks, by means of different new constraints based on MMD and Sinkhorn Divergences. We derive learning bounds establishing that the learned representation transfers well to novel tasks. We present experiments on three real world datasets, showing that the proposed method outperforms state-of-the-art approaches by a significant margin.

## 1 Introduction

During the last decade, the widespread distribution of automatic systems for decision making is raising concerns about their potential for unfair behaviour [3, 7, 37]. As a consequence, machine learning models have often to meet fairness requirements, ensuring the correction and limitation of – for example – racist or sexist decisions. In literature, it is possible to find a plethora of different methods to generate fair models with respect to one or more sensitive attributes (e.g. gender, ethnic group, age). These methods can be mainly divided into three families: (i) methods in the first family change a pre-trained model in order to make it more fair (while trying to maintain the classification performance, i.e., post-processing of the model) [14, 19, 34]; (ii) in the second family, we can find methods that enforce fairness directly during the training phase, e.g. [1, 12, 41, 42]; (iii) the third family of methods implements fairness by modifying the data representation (i.e., pre-processing of the data), and then employs standard machine learning methods [9, 43].

All methods in the previous three families have in common the goal of creating a fair and accurate model from scratch on the specific task at hand. This solution may work well in specific cases, but in a large number of real world applications, using the same model (or at least part of it) over different tasks is helpful if not mandatory. For example, it is common to perform a fine tuning over pre-trained models [11], keeping fixed the internal representation. Indeed, most modern machine learning frameworks (especially the deep learning ones) offer a set of pre-trained models that are distributed in

so-called model zoos[1]. Unfortunately, fine tuning pre-trained models on novel previously unseen tasks could lead to an unexpected unfairness behaviour, even starting from an apparently fair model for previous tasks (e.g. discriminatory transfer [24] or negative legacy [22]), due to missing generalization guarantees concerning the fairness property of the model over new tasks.

In order to overcome the above problem, in this paper we follow the framework of multitask learning. We aim to leverage task similarities to learn a fair representation that provably generalizes well to unseen tasks. By this we mean that when the representation is used to learn novel tasks, it is guaranteed to learn a model that has both a small error and meets the fairness requirements. We measure fairness according to demographic parity [8] (for an extended analysis of different fairness definitions see [39, 42]). It requires the probability of possible model decisions to be independent of the sensitive information. We argue that multitask methods are well suited to learn a shared fair representation according to demographic parity. The fairness of the representation is enforced by imposing the distributions of the different subgroups to be close with respect to three different distances, respectively between their: (i) average value (AVG) [32], (ii) Maximum Mean Discrepancy (MMD) [35, 38], and (iii) Sinkhorn Divergences (SNK) [10].

**Contributions.** We propose a method for learning a shared fair representation across the multiple tasks, by incorporating novel constraints based on MMD and Sinkhorn Divergences on the representation. We show empirically and theoretically that the representation learned by the method transfers well to novel tasks in the sense that whenever the empirical unfairness is small on the training tasks then the unfairness on a future task will likely be small as well. An important implication of our results is that the learned representation can be used *as is* to learn models for new unseen tasks, that are provably fair, without the need of imposing any further fairness constraint on the model.

**Organization.** The paper is organized as follows. In Section 2, we discuss previous related work aimed at learning fair representations. In Section 3, we introduce the proposed method. In Section 4, we study the generalization properties of our method. In Section 5, we experimentally compare the proposed method against different baselines and state-of-the-art approaches on three real world datasets. Finally, in Section 6 we discuss directions of future research.

## 2    Related work

In this work – and more generally in the current literature [6, 13, 21, 25, 26, 30, 31, 40, 43] – with fair representation we refer to the concept of learning a function mapping the raw input into a set of features that do not discriminate subgroups in the data, in the sense that the transformed input is conditionally independent of subgroup membership. This approach is different from most commonly used approaches [12, 19, 41], in which the focus is to solve a task (or a set of tasks) without discriminating subgroups in the data, regardless of the fairness of the representation itself. In the previously mentioned papers, a fair model is learned directly from the raw data, without performing any explicit representation extraction.

In particular, in [6, 13, 25, 26, 30, 31, 40], the authors propose different neural networks (NN) architectures together with modified learning strategies able to learn a representation that obscures or removes the sensitive variable. In the general case, all these methods have an input, a target variable (i.e. the task at hand) and a binary sensitive variable. The objective is to learn a representation that: (i) preserves information about the input space; (ii) is useful for predicting the target; (iii) is approximately independent of the sensitive variable. In practice, these methods pursue the goal of making the generated model act randomly when the internal representation is exploited to predict the sensitive variable. In this sense, no actual constraint is directly imposed on the internal representation, but only on the output of the model.

A different direction is taken in [21], where the authors show how to formulate the problem of counterfactual inference as a domain adaptation problem, and more specifically a covariate shift problem [36]. The authors derive two new families of representation algorithms for counterfactual inference. The first one is based on linear models and variable selection, and the other one on deep learning. The authors show that learning representations that encourage similarity (i.e., balance) between the treatment and control populations leads to better counterfactual inference; this is in contrast to many methods which attempt to create balance by re-weighting samples.

Finally, in [43], the authors learn a representation of the data that is a probability distribution over clusters, where the cluster of a datapoint contains no-information about the sensitive variable, namely

fair clustering. In this sense, the clustering is learned to be fair and also discriminative for the prediction task at hand.

## 3 Method

In this section, we present our method to learn a shared fair representation within a general multitask learning setting. We consider $T$ supervised learning tasks (i.e. binary classification or regression problems). Each task $t \in \{1, \ldots, T\}$ is identified by a probability distribution $\mu_t$ on $\mathcal{X} \times \mathcal{S} \times \mathcal{Y}$, where $\mathcal{X} \subset \mathbb{R}^d$ is the set of non-sensitive input variables, $\mathcal{S} = \{1, 2\}$ is the set of values of a binary sensitive variable[2] and $\mathcal{Y}$ is the output space, which is either $\{-1, 1\}$ for binary classification or $\mathcal{Y} \subset \mathbb{R}$ for regression. We let $\mathcal{D}_t = (x_{t,i}, s_{t,i}, y_{t,i})_{i=1}^m \in (\mathcal{X} \times \mathcal{S} \times \mathcal{Y})^m$ be the training sequence for task $t$, which is sampled independently from $\mu_t$. For each $s \in \{1, 2\}$ we also let $\mathbf{x}_t = (x_{t,i} : s_i = 1)$ and $\mathbf{z}_t = (x_{t,i} : s_i = 2)$ be the set of inputs in the first and second group, respectively. We consider compositional models with a shared representation, that is $f_t(x) = g_t(h(x))$, where for a prescribed positive integer $r$,

$$h : \mathcal{X} \to \mathbb{R}^r$$

is a representation function and the functions $g_t : \mathbb{R}^r \to \mathcal{Y}$ are task specific[3].

We require the model to satisfy the demographic parity fairness constraint [16, 39] *at the representation level*. That is, we demand that the conditional distribution of $h(x)$ is the same across the two subgroups. Formally this means, for every measurable subset $C \subset \mathbb{R}^r$, and for every task $t \in \{1, \ldots, T\}$, that

$$\text{Prob}(h(X) \in C \,|S = 1) = \text{Prob}(h(X) \in C \,|S = 2). \tag{1}$$

Notice that if demographic parity is satisfied at the representation level – i.e., Eq. (1) holds true – then every model built from such representation will satisfy demographic parity as well, that is, the distribution of the predicted output is the same for each of the subgroups. In the next section we will show that, if the tasks are randomly observed, then demographic parity will also be satisfied on future tasks with high probability. In this sense our method can be interpreted as learning a fair transferable representation.

The constraint (1) is difficult to handle, therefore we relax it by requiring that for every $t \in \{1, \ldots, T\}$, the corresponding distributions are close to each other according to a suitable metric on probability distributions $d : \mathcal{P}(\mathcal{X}) \times \mathcal{P}(\mathcal{X})$, where $\mathcal{P}(\mathcal{X})$ is the set of probability measures on $\mathcal{X}$. We consider two well established metrics, maximum mean discrepancy (MMD) and Sinkhorn divergence.

**Maximum mean discrepancy.** Let $K : \mathcal{X} \times \mathcal{X} \to \mathbb{R}$ be a positive definite kernel and let $\Psi : \mathcal{X} \to \mathbb{H}$ a corresponding feature map, that is, for every $x, y \in \mathcal{X}$, we have $K(x, y) = \langle \Psi(x), \Psi(y) \rangle_{\mathbb{H}}$, where $\mathbb{H}$ is a Hilbert space with inner product $\langle \cdot, \cdot \rangle_{\mathbb{H}}$. If $P, Q \in \mathcal{P}(\mathcal{X})$, their squared maximum mean discrepancy (MMD$^2$) relative to the kernel $K$ is defined as

$$\text{MMD}^2(P, Q) = \| \mathbb{E}_{X \sim Q} \Psi(X) - \mathbb{E}_{X \sim Q} \Psi(X) \|_{\mathbb{H}}^2. \tag{2}$$

Moreover if $\mathbf{x} = (x_i)_{i=1}^n$ and $\mathbf{z} = (z_i)_{i=1}^m$ are two independent samples from $P$ and $Q$, respectively, their MMD$^2$ is defined as the MMD$^2$ between the corresponding empirical distributions $\hat{P} = \frac{1}{n} \sum_{i=1}^n \delta(x_i - \cdot)$ and $\hat{Q} = \frac{1}{m} \sum_{i=1}^m \delta(z_i - \cdot)$. This (V-statistic) estimator has a bias of order $O(1/\min(n, m))$. A slightly different unbiased estimator is given by[4]

$$\text{MMD}^2(\hat{P}, \hat{Q}) = \frac{1}{n(n-1)} \sum_{i \neq j} K(x_i, x_j) + \frac{1}{m(m-1)} \sum_{i \neq j} K(z_i, z_j) - \frac{2}{nm} \sum_{i,j} K(x_i, z_j). \tag{3}$$

Our experiments below use this estimator.

**Sinkhorn divergence.** For any $P, Q \in \mathcal{P}(\mathcal{X})$, the Optimal Transport problem with entropic regularization is defined as [33]

$$\text{OT}_\varepsilon(P, Q) = \min_{\pi \in \Pi(P,Q)} \int_{\mathcal{X}^2} \|x - y\|^2 \, d\pi(x, y) + \varepsilon\text{KL}(\pi | P \otimes Q), \qquad \varepsilon \geq 0 \qquad (4)$$

where $\text{KL}(\pi | P \otimes Q)$ is the *Kullback-Leibler divergence* between the candidate transport plan $\pi$ and the product distribution $P \otimes Q$, and $\Pi(P, Q) = \{\pi \in \mathcal{P}(\mathcal{X} \times \mathcal{X}) \colon \pi_1 = P, \ \pi_2 = Q\}$, with $\pi_1$ and $\pi_2$ the marginals of $\pi$. The case $\varepsilon = 0$ corresponds to the classic Optimal Transport problem introduced by Kantorovich [23]. Sinkhorn divergence is defined as

$$\text{S}_\varepsilon(P, Q) = \text{OT}_\varepsilon(P, Q) - \frac{1}{2}\text{OT}_\varepsilon(P, P) - \frac{1}{2}\text{OT}_\varepsilon(Q, Q) \qquad (5)$$

and was shown in [15] to be nonnegative, biconvex and to metrize the convergence in law under mild assumptions.

Below, for every $\mathbf{x} \sim P^n$ and $\mathbf{z} \sim Q^n$, with some abuse of notation, we denote with $d(\mathbf{x}, \mathbf{z})$ either the MMD$^2$ or Sinkhorn divergence estimator. Furthermore, we use the notation $h(\mathbf{x}) = (h(x_i))_{i=1}^n$ and $h(\mathbf{z}) = (h(z_i))_{i=1}^n$, so that $d(h(\mathbf{x}), h(\mathbf{z}))$ is the MMD$^2$ or Sinkhorn divergence estimator of the transformed samples.

### 3.1 Algorithm

Our method is based on regularized empirical risk minimization, in which the empirical risk is an average multitask objective combining a prediction error term and an unfairness term. Specifically, we consider the problem

$$\min_{h \in \mathcal{H}, g_1, \ldots, g_T \in \mathcal{G}} \quad \frac{1}{T} \sum_{t=1}^T \left\{ \sum_{i=1}^m \ell\big(y_{t,i}, g_t(h(x_{t,i}))\big) + \gamma d\big(h(\mathbf{x}_t), h(\mathbf{z}_t)\big) \right\} \qquad (6)$$

where $\ell(\cdot, \cdot)$ is a loss function, e.g. the squared loss or logistic, and $\gamma$ is positive parameter trading off the desiderata of having small error and small unfairness.

The optimization in (6) is over classes $\mathcal{H}$ and $\mathcal{G}$ of possible representations and task specific functions. In our empirical study below, we focus on 1-hidden layer networks models, that is we choose

$$h(x) = \sigma(Wx) \qquad (7)$$

where $W$ is an $r \times d$ matrix of bounded Frobenious norm, $\sigma : \mathbb{R} \to \mathbb{R}$ is an activation function (e.g., sigmoid) and $g_t$ are linear functions, that is $g_t(\cdot) = \langle v_t, \cdot \rangle$, with $v_t \in \mathbb{R}^r$ a vector of parameters of bounded euclidean norm. The corresponding version of (6), written in an easier to optimize unconstrained way, is

$$\min_{W, V} \quad \frac{1}{T} \sum_{t=1}^T \left\{ \sum_{i=1}^m \ell\big(y_{t,i}, \langle v_t, \sigma(Wx_{t,i})\rangle\big) + \gamma d(\sigma(W\mathbf{x}_t), \sigma(W\mathbf{z}_t)) \right\} + \lambda\big(\|W\|_F^2 + \|V\|_F^2\big) \quad (8)$$

where $W \in \mathbb{R}^{d \times r}$, $V = [v_1 \ldots v_T] \in \mathbb{R}^{r \times T}$, $\| \cdot \|_F$ is the Frobenius norm, and $\lambda$ is a positive regularization parameter. We solve Problem (8) by gradient descent. For the fairness measure, we optimize Sinkhorn divergence using automatic differentiation [17], while for MMD$^2$ the computation is direct by the chain rule (assuming the kernel to be differentiable).

## 4 Learning bounds for MMD

In this section we present learning bounds for the proposed method. We focus on fairness guarantees, since risk bounds are well established, see, e.g. [5, 28, 29] and references therein. Particularly, bounds for 1-hidden layer networks of the form considered here, are presented in [29, Thm. 5].

We consider the setting of learning-to-learn [5], in which the tasks are random realizations from a meta-distribution $\rho$ over the set of possible tasks (also called the *environment* in the learning-to-learn literature). For our purpose it is enough to regard a task as a pair of distributions $(P, Q) \in \mathcal{P}(\mathcal{X}) \times \mathcal{P}(\mathcal{X})$ associated to the two sensitive groups. For simplicity in our analysis we assume that we draw samples of equal size form each distribution.[5]

Let $h : \mathcal{X} \to \mathbb{R}^r$. For any probability measure $P \in \mathcal{P}(\mathcal{X})$, the pushforward measure of $P$ via $h$ is the probability measure $h_\# P \in \mathcal{P}(\mathbb{R}^r)$ defined for any Borel subset $V$ or $\mathbb{R}^r$ as $(h_\# P)(V) = P(h^{-1}(V))$. Our goal is to bound the unfairness of the representation $\bar{h}$ found by solving problem (8) on a future random task in terms of the average empirical unfairness on the training tasks, that is

$$\mathbb{E}_{(P,Q)\sim\rho} d(\bar{h}_\sharp P, \bar{h}_\sharp Q) \leq \frac{1}{T} \sum_{t=1}^{T} d(\bar{h}(\mathbf{x}_t), \bar{h}(\mathbf{z}_t)) + \mathrm{Gap}(T, n, \delta).$$

Notice the the l.h.s. in the above bound measures the average unfairness of the representation at the population level, that is using the true distributions of a task rather that their empirical counterpart. This is the quantity that we wish to be small in order for the demographic parity constraint (1) to be approximately satisfied. The bound holds with probability larger than $1 - \delta$ in the draw of the training tasks and their samples, where $\delta \in (0, 1)$ is a small confidence parameter that increases the bound only logarithmically. Since the multitask empirical unfairness of the representation $\bar{h}$ found by our method is expected to be small, if $\mathrm{Gap}(T, n, \delta)$ decreases in $n$ and $T$ then the bound guarantees that the unfairness of $\bar{h}$ on future tasks will likely remain small.

Of course $\bar{h}$ is not known in advance, so we bound the uniform deviation between expected and empirical multitask unfairness over the class of possible representations $\mathcal{H}$. Our bound is expressed in terms of the Rademacher average of the set $\mathcal{H}(\mathbf{X}, \mathbf{Z})) = \{(h(\mathbf{x}_1), h(\mathbf{z}_1), \ldots, h(\mathbf{x}_t), h(\mathbf{z}_t)) : h \in \mathcal{H}\} \subseteq \mathbb{R}^{2nrT}$, which is defined as

$$R(\mathcal{H}(\mathbf{X}, \mathbf{Z})) = \mathbb{E}_\epsilon \sup_{h \in \mathcal{H}} \sum_{t=1}^{T} \sum_{i=1}^{n} \sum_{k=1}^{r} \left\{ \epsilon_{t,i,k} h_k(x_{t,i}) + \epsilon'_{t,i,k} h_k(z_{t,i}) \right\} \tag{9}$$

where $\epsilon_{t,i,k}$ and $\epsilon'_{t,i,k}$ are i.i.d. Rademacher random variables.

**Theorem 1.** *Let $d$ be the unbiased $\mathrm{MMD}^2$ estimator (3). Let $(P_1, Q_1), \ldots, (P_T, Q_T)$ be independently sampled from $\rho$ and, for every $t \in \{1, \ldots, T\}$, let $\mathbf{x}_t \sim P_t^n$ and $\mathbf{z}_t \sim Q_t^n$. Then it holds with probability at least $1 - \delta$ in the draw of the multi-sample $(\mathbf{X}, \mathbf{Z}) = (\mathbf{x}_t, \mathbf{z}_t)_{t=1}^{T}$, that*

$$\sup_{h \in \mathcal{H}} \left\{ \mathbb{E}_{(P,Q)\sim\rho} d(h_\sharp P, h_\sharp Q) - \frac{1}{T} \sum_{t=1}^{T} d(h(\mathbf{x}_t), h(\mathbf{z}_t)) \right\} \leq C_1 \frac{L_d}{T} R(\mathcal{H}(\mathbf{X}, \mathbf{Z})) + C_2 \sqrt{\frac{\log 1/\delta}{T}} \tag{10}$$

*where $L_d$ is the Lipschitz constant of the mapping $(\mathbf{x}, \mathbf{z}) \mapsto d(\mathbf{x}, \mathbf{z})$ w.r.t. the Euclidean norm and $C_1$ and $C_2$ are small positive numerical constants.*

**Proof sketch.** We add and subtract the term $\mathbb{E}_{(P,Q)\sim\rho} \mathbb{E}_{(\mathbf{x},\mathbf{z})\sim(P\times Q)^n} d(h(\mathbf{x}), h(\mathbf{z}))$ inside the supremum in the l.h.s. of (10) and decompose it as the sum of two terms, $\sup_h A_h + \sup_h B_h$, where

$$A_h = \mathbb{E}_{(P,Q)\sim\rho} \mathbb{E}_{(\mathbf{x},\mathbf{z})\sim(P\times Q)^n} [d(h_\# P, h_\# Q) - d(h(\mathbf{x}), h(\mathbf{z}))] \text{ and}$$

$$B_h = \mathbb{E}_{(P,Q)\sim\rho} \mathbb{E}_{(\mathbf{x},\mathbf{z})\sim(P\times Q)^n} d(h(\mathbf{x}), h(\mathbf{z})) - \frac{1}{T} \sum_{t=1}^{T} d(h(\mathbf{x}_t), h(\mathbf{z}_t)).$$

For a fixed $h$, the quantity $A_h$ measures the average bias of the estimator over the draw of the distributions from the environment. For the $\mathrm{MMD}^2$ estimator (3) this term is zero. The term $\sup_h B_h$ can be bounded with a uniform Rademacher bound, with leading term

$$\frac{2}{T} \mathbb{E}_\varepsilon \sup_{h \in \mathcal{H}} \sum_{t=1}^{T} \epsilon_t d(h(\mathbf{x}_t), h(\mathbf{z}_t)) \tag{11}$$

plus the last term in the r.h.s. of (10), where $\epsilon_1, \ldots, \epsilon_T$ are i.i.d. Rademacher. We then appeal to [27, Ineq. (1)] to bound (11) by $L_d$ times the Rademacher average of the set $\mathcal{H}(\mathbf{X}, \mathbf{Z})$. ∎

We note that the same proof of Theorem 1 applies to the V-statistic $\mathrm{MMD}^2$ estimator mentioned after equation (3). In this case the bias term $A$ is of order $O(1/n)$, whereas the Lipschitz constant remains the same – see the discussion in the appendix. On the other hand in order to extent Theorem 1 to Sinkhorn divergence, we would need to find a formula for both the bias term and the Lipschitz constant of the empirical Sinkhorn divergence w.r.t. the euclidean norm.

In the case of linear representations composed with an activation function in Eq. (7) and matrix $W$ with bounded Frobenius norm (below for simplicity bounded by 1) the Rademacher average is of order $L_\sigma \sqrt{rnT}$ where $L_\sigma$ is the Lipschitz constant of the activation function.

**Corollary 2.** *Consider the class of representations* (7)*, assume that the activation function is Lipschitz with constant $L_\sigma$ and that the matrix $W$ of parameters has Frobenius norm bounded by* 1*. Assume further that $\|x\| \leq B$ a.s. over the environment $\rho$. Then under the same assumptions in Theorem 1 it holds that*

$$\mathbb{E}_{(P,Q)\sim\rho} d(h_\sharp P, h_\sharp Q) - \frac{1}{T}\sum_{t=1}^{T} d(h(\mathbf{x}_t), h(\mathbf{z}_t)) \leq C_1 L_\sigma B\sqrt{\frac{r}{T}} + C_2\sqrt{\frac{\log 1/\delta}{T}}.$$

The proof uses standard techniques for bounding the Rademacher average of linear function classes, see Proposition 5 together with Lemma 4, bounding the Lipschitz constant of the MMD$^2$ estimator by $O(1/\sqrt{n})$. Both results can be found in the appendix. The above corollary tell us that if $T$ is significantly greater than the size of the representation then the empirical MMD on the training tasks predicts well the MMD on a new task, on average. On the contrary if the representation is learned independently on any new task the generalization bound will be of order $\sqrt{\frac{r}{n}}$ which may be much larger than the order of $\sqrt{\frac{r}{T}}$ for our method, since in practical applications $T$ may be much larger than $n$.

At last, now that we have bounded the MMD at the representation level, we discuss how we can then derive a bound on the fairness of at the output level.

**Lemma 3.** *Let $\bar{h}$ be the representation learned by method* (8) *and let with some abuse of notation $\bar{P}, \bar{Q} \in \mathcal{P}(\mathbb{R}^r)$ be the probability distributions associated to the two sensitive groups after the raw input is preprocessed with $\bar{h}$. Let the* MMD *distance at the representation level be measured w.r.t. kernel $K : \mathbb{R}^r \times \mathbb{R}^r \to \mathbb{R}$ and the* MMD *on the output be measured w.r.t. kernel $G : \mathbb{R} \times \mathbb{R} \to \mathbb{R}$. Then, for every $v \in \mathbb{R}^r$ we have*

$$\text{MMD}_G(v_\#\bar{P}, v_\#\bar{Q}) \leq \sup_{\|g\|_G \leq 1} \|g(\langle v, \cdot\rangle)\|_K \text{MMD}_K(\bar{P}, \bar{Q}).$$

This result shows that if we have a class of possible output weight vectors $v$, say a ball of radius 1, then provided $\sup_{\|v\|\leq 1} \sup_{\|g\|_G \leq 1} \|g(\langle v, \cdot\rangle)\|_K$ is bounded, then the MMD at the representation level controls the unfairness of any regression or classification function used on top of this representation.

## 5 Empirical Study

In this section, we compare our proposal against different baselines and state-of-the-art techniques.

**Setting.** To study the performance of our method, we perform two sets of experiments, one in the linear setting and one in the non-linear setting. The first set of experiments (Table 1 and Figure 1) compares: a single layered feed-forward NN (FFNN) with linear activation and no fairness constraints (UNC), constraining the output of each task with [12] (M [12]), the fair shared representation methods presented in [26] (M [26]) and in [13] (M [13]) – both providing also the code – and the fair shared representation proposed in this work using three different constraints: Mean Matching $M_{\text{AVG}}$ (i.e. MMD with linear kernel), Maximum Mean Discrepancy with Gaussian kernel $M_{\text{MMD}}$, and Sinkhorn Divergence $M_{\text{SNK}}$. In the second set of experiments (Table 2 and Figure 2), we report the equivalent results of Table 1 and Figure 1 when, in the representation layer, a sigmoidal non-linear activation functions is added. This new table represents the non-linear scenario.

We test each method either on the same tasks exploited during the training phase, or on novel tasks. Concerning the experiments on the same task setting, we train the model with all the tasks and then we measure results on an independent test set of the same tasks. In the case of novel task experiments, we train the model with all the tasks minus one (randomly selected). Then, we fix the representation and we use a subset of the data (70%) of the excluded task to train the last layer, maintaining fixed the representation layer. The remaining data (30%) of the novel task is used to measure error and fairness. We consider both the case where the sensitive feature is present, or not in the functional form of the model (i.e., the sensitive feature is known or not in the testing phase).

We validate the hyperparameters using a grid search with $\lambda \in \{10^{-6.0}, 10^{-5.8}, \ldots, 10^{+4.0}\}$, $\gamma \in \{10^{-1.0}, 10^{-0.5}, \ldots, 10^2\}$, and $r \in \{2^j d \mid j = -4, -3, \ldots, 10\}$, following the validation procedure in [12]. Firstly, the classical 10-fold CV error for each of the combination of the hyperparameters is computed. Then, we shortlist all the hyperparameters' combinations with error close to the best one (above 90% of the smallest error). From this list, we select the hyperparameters with the smallest fairness risk. Concerning the error (ERR) we used the percentage of misclassifications, and concerning the fairness measure of our model (DDP), we compute the absolute value of the

Table 1: Feed Forward Single Layered NN with Linear Activation Functions. Comparison of (UNC) no fairness constraints, (M [12]) constraining the output fairness of each task with [12], (M [26]) the fair shared representation method in [26], (M [13]) the fair shared representation method in [13], and the fair shared representation proposed in this work using different constraints (Mean Matching $M_{AVG}$, Maximum Mean Discrepancy $M_{MMD}$, and Sinkhorn Divergence $M_{SNK}$).

| | | UNC | | M [12] | | M [26] | | M [13] | | $M_{AVG}$ | | $M_{MMD}$ | | $M_{SNK}$ | |
|---|---|---|---|---|---|---|---|---|---|---|---|---|---|---|---|
| | Data | ERR | DDP | ERR | DDP | ERR | DDP | ERR | DDP | ERR | DDP | ERR | DDP | ERR | DDP |
| | | **Sensitive feature not in the functional form of the model** | | | | | | | | | | | | | |
| Same Tasks | SCH | $10.7^{\pm.6}$ | $.077^{\pm.003}$ | $12.3^{\pm.8}$ | $.013^{\pm.001}$ | $13.4^{\pm1.0}$ | $.017^{\pm.002}$ | $12.9^{\pm.8}$ | $.018^{\pm.002}$ | $11.8^{\pm.8}$ | $.011^{\pm.001}$ | $11.9^{\pm.7}$ | $.009^{\pm.002}$ | $11.5^{\pm.7}$ | $.008^{\pm.001}$ |
| | UNI | $13.7^{\pm.5}$ | $.070^{\pm.003}$ | $18.1^{\pm.9}$ | $.012^{\pm.001}$ | $21.2^{\pm1.3}$ | $.021^{\pm.004}$ | $26.2^{\pm2.}$ | $.027^{\pm.004}$ | $15.0^{\pm.5}$ | $.010^{\pm.001}$ | $14.3^{\pm.6}$ | $.009^{\pm.001}$ | $15.4^{\pm.5}$ | $.008^{\pm.001}$ |
| | MOV | $15.1^{\pm.6}$ | $.112^{\pm.008}$ | $17.1^{\pm.7}$ | $.009^{\pm.001}$ | $19.2^{\pm0.9}$ | $.014^{\pm.002}$ | $18.0^{\pm.8}$ | $.012^{\pm.002}$ | $17.3^{\pm.8}$ | $.007^{\pm.001}$ | $16.6^{\pm.4}$ | $.005^{\pm.002}$ | $16.9^{\pm.7}$ | $.005^{\pm.002}$ |
| | | **Sensitive feature in the functional form of the model** | | | | | | | | | | | | | |
| | SCH | $9.6^{\pm.4}$ | $.085^{\pm.004}$ | $11.0^{\pm.9}$ | $.020^{\pm.001}$ | $12.0^{\pm1.0}$ | $.022^{\pm.002}$ | $13.3^{\pm1.}$ | $.025^{\pm.002}$ | $10.7^{\pm.5}$ | $.019^{\pm.001}$ | $10.5^{\pm.6}$ | $.013^{\pm.002}$ | $10.2^{\pm.5}$ | $.014^{\pm.002}$ |
| | UNI | $12.3^{\pm.7}$ | $.077^{\pm.004}$ | $13.8^{\pm.8}$ | $.017^{\pm.001}$ | $20.1^{\pm1.2}$ | $.029^{\pm.005}$ | $25.9^{\pm2.}$ | $.032^{\pm.006}$ | $13.7^{\pm.8}$ | $.017^{\pm.001}$ | $13.2^{\pm.7}$ | $.013^{\pm.001}$ | $13.9^{\pm.9}$ | $.017^{\pm.001}$ |
| | MOV | $13.0^{\pm.5}$ | $.123^{\pm.007}$ | $15.7^{\pm.7}$ | $.010^{\pm.001}$ | $18.9^{\pm0.7}$ | $.017^{\pm.004}$ | $17.1^{\pm.9}$ | $.015^{\pm.003}$ | $15.2^{\pm.7}$ | $.011^{\pm.001}$ | $15.1^{\pm.6}$ | $.009^{\pm.001}$ | $14.3^{\pm.8}$ | $.008^{\pm.002}$ |
| | | **Sensitive feature not in the functional form of the model** | | | | | | | | | | | | | |
| New Tasks | SCH | $13.8^{\pm.5}$ | $.088^{\pm.003}$ | $15.6^{\pm0.8}$ | $.032^{\pm.002}$ | $16.4^{\pm1.1}$ | $.044^{\pm.004}$ | $17.2^{\pm1.}$ | $.041^{\pm.004}$ | $14.8^{\pm.7}$ | $.022^{\pm.001}$ | $14.9^{\pm.8}$ | $.020^{\pm.002}$ | $14.8^{\pm.6}$ | $.017^{\pm.001}$ |
| | UNI | $15.6^{\pm.8}$ | $.075^{\pm.003}$ | $16.2^{\pm0.9}$ | $.021^{\pm.002}$ | $22.0^{\pm1.5}$ | $.029^{\pm.004}$ | $27.3^{\pm1.}$ | $.033^{\pm.005}$ | $17.0^{\pm.7}$ | $.015^{\pm.001}$ | $16.5^{\pm.7}$ | $.011^{\pm.001}$ | $16.0^{\pm.6}$ | $.009^{\pm.002}$ |
| | MOV | $18.2^{\pm.8}$ | $.128^{\pm.007}$ | $19.2^{\pm0.9}$ | $.025^{\pm.002}$ | $21.2^{\pm1.4}$ | $.031^{\pm.004}$ | $20.1^{\pm1.}$ | $.030^{\pm.003}$ | $20.3^{\pm1.}$ | $.016^{\pm.001}$ | $20.9^{\pm.9}$ | $.016^{\pm.001}$ | $18.9^{\pm.8}$ | $.011^{\pm.001}$ |
| | | **Sensitive feature in the functional form of the model** | | | | | | | | | | | | | |
| | SCH | $12.7^{\pm.5}$ | $.096^{\pm.005}$ | $14.7^{\pm0.9}$ | $.038^{\pm.002}$ | $18.0^{\pm1.1}$ | $.042^{\pm.003}$ | $17.9^{\pm.9}$ | $.056^{\pm.003}$ | $13.8^{\pm.8}$ | $.030^{\pm.002}$ | $13.5^{\pm.8}$ | $.024^{\pm.002}$ | $13.1^{\pm.7}$ | $.018^{\pm.002}$ |
| | UNI | $14.2^{\pm.7}$ | $.082^{\pm.001}$ | $15.9^{\pm0.7}$ | $.029^{\pm.002}$ | $19.2^{\pm1.0}$ | $.035^{\pm.005}$ | $25.9^{\pm1.}$ | $.038^{\pm.006}$ | $15.6^{\pm.6}$ | $.022^{\pm.001}$ | $15.1^{\pm.6}$ | $.017^{\pm.001}$ | $15.0^{\pm.6}$ | $.017^{\pm.001}$ |
| | MOV | $16.1^{\pm.9}$ | $.139^{\pm.011}$ | $20.1^{\pm0.7}$ | $.038^{\pm.002}$ | $20.1^{\pm1.1}$ | $.037^{\pm.003}$ | $19.9^{\pm1.}$ | $.038^{\pm.004}$ | $18.2^{\pm.8}$ | $.027^{\pm.001}$ | $18.0^{\pm.7}$ | $.018^{\pm.002}$ | $18.1^{\pm.8}$ | $.020^{\pm.001}$ |

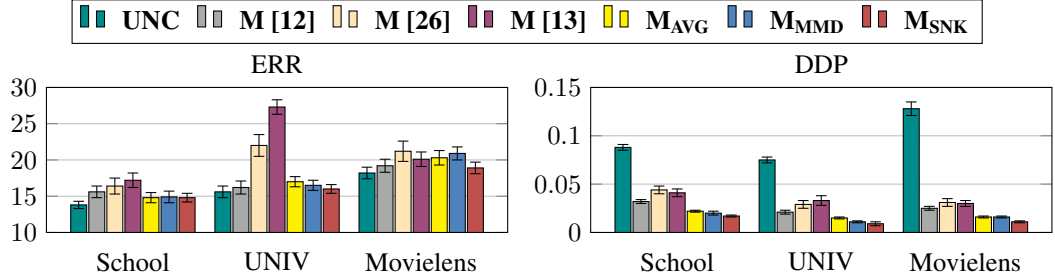

Figure 1: Graphical representation of the results in Table 1 for new tasks when the sensitive feature is not included in the functional form of the model.

difference of demographic parity as $\frac{1}{|\mathcal{Y}|} \sum_{y \in \mathcal{Y}} |\mathcal{P}(f(x(,s)) = y|s = 1) - \mathcal{P}(f(x(,s)) = y|s = 2)|$, since in our datasets the output space is finite. For all the experiments, we report performance over 30 repetitions with standard deviation.

**Datasets.** In our comparisons we used three datasets. The first one is the School dataset [18] (SCH) – by the Inner London Education Authority (ILEA) – with examination records of 15362 students from 139 secondary schools in years $1985 - 1987$. The goal is to predict exam scores for students in one school, based on eight inputs. Four inputs (year of the exam, gender, VR band, and ethnic group) are student-dependent, and four others (percentage of students eligible for free school meals, percentage of students in VR band one, school gender – mixed or single-gender – and school denomination) are school-dependent. The categorical variables were one-hot encoded, making a total of 22 inputs.We scaled each covariate and output to have zero mean and unit variance. The second dataset we propose has been collected at the University of Genoa[6] (UNI). This dataset is a proprietary and highly sensitive dataset containing all the data about the past and present students enrolled at the UNIV. In this study we take into consideration students who enrolled in the academic year 2017-2018. The dataset contains 5000 instances with 35 attributes each (both numeric and categorical) about ethnicity, gender, financial status, and previous school experience. The tasks are to predict the grades of the 10 exams of the first semester. Finally, the third dataset is Movielens [20] (MOV) – specifically Movielens 100k (ml100k) – which consists of ratings (1 to 5) provided by 943 users for a set of 1682 movies, with 100,000 ratings available. The tasks are to predict the preference of a user over the movies. Additional features for each movie, such as the year of release or its genre, are provided. In all datasets, the sensitive attribute is the gender.

Table 2: Feed Forward Single Layered NN with Sigmoidal Activation Functions. Comparison of (UNC) no fairness constraints, (M [12]) constraining the output fairness of each task with [12], (M [26]) the fair shared representation method in [26], (M [13]) the fair shared representation method in [13], and the fair shared representation proposed in this work using different constraints (Mean Matching $M_{AVG}$, Maximum Mean Discrepancy $M_{MMD}$, and Sinkhorn Divergence $M_{SNK}$).

| | | UNC | | M [12] | | M [26] | | M [13] | | $M_{AVG}$ | | $M_{MMD}$ | | $M_{SNK}$ | |
|---|---|---|---|---|---|---|---|---|---|---|---|---|---|---|---|
| | Data | ERR | DDP | ERR | DDP | ERR | DDP | ERR | DDP | ERR | DDP | ERR | DDP | ERR | DDP |
| | | | | | | Sensitive feature not in the functional form of the model | | | | | | | | | |
| Same Tasks | SCH | $6.8^{\pm.8}$ | $.068^{\pm.013}$ | $9.9^{\pm.3}$ | $.015^{\pm.001}$ | $10.1^{\pm.4}$ | $.016^{\pm.001}$ | $9.9^{\pm.4}$ | $.010^{\pm.001}$ | $7.9^{\pm.4}$ | $.009^{\pm.001}$ | $7.5^{\pm.5}$ | $.007^{\pm.003}$ | $7.1^{\pm.5}$ | $.006^{\pm.003}$ |
| | UNI | $8.9^{\pm.7}$ | $.151^{\pm.003}$ | $10.9^{\pm.6}$ | $.091^{\pm.005}$ | $11.1^{\pm.6}$ | $.099^{\pm.006}$ | $12.4^{\pm.5}$ | $.101^{\pm.007}$ | $10.4^{\pm.6}$ | $.091^{\pm.005}$ | $10.2^{\pm.6}$ | $.072^{\pm.007}$ | $10.0^{\pm.6}$ | $.074^{\pm.005}$ |
| | MOV | $7.7^{\pm.8}$ | $.091^{\pm.008}$ | $9.1^{\pm.4}$ | $.001^{\pm.001}$ | $9.4^{\pm.4}$ | $.002^{\pm.002}$ | $9.9^{\pm.4}$ | $.003^{\pm.002}$ | $8.7^{\pm.4}$ | $.001^{\pm.001}$ | $8.3^{\pm.7}$ | $.002^{\pm.001}$ | $8.9^{\pm.6}$ | $.004^{\pm.002}$ |
| | | | | | | Sensitive feature in the functional form of the model | | | | | | | | | |
| | SCH | $6.6^{\pm.6}$ | $.073^{\pm.004}$ | $9.4^{\pm.3}$ | $.021^{\pm.001}$ | $9.2^{\pm.5}$ | $.019^{\pm.003}$ | $8.9^{\pm.4}$ | $.019^{\pm.002}$ | $7.2^{\pm.4}$ | $.015^{\pm.001}$ | $7.3^{\pm.4}$ | $.011^{\pm.003}$ | $7.1^{\pm.6}$ | $.009^{\pm.001}$ |
| | UNI | $8.8^{\pm.7}$ | $.197^{\pm.004}$ | $9.2^{\pm.4}$ | $.120^{\pm.011}$ | $10.7^{\pm.4}$ | $.154^{\pm.014}$ | $10.3^{\pm.4}$ | $.161^{\pm.013}$ | $9.5^{\pm.4}$ | $.155^{\pm.010}$ | $9.3^{\pm.5}$ | $.117^{\pm.010}$ | $9.1^{\pm.5}$ | $.088^{\pm.008}$ |
| | MOV | $7.6^{\pm.7}$ | $.101^{\pm.007}$ | $8.1^{\pm.3}$ | $.009^{\pm.001}$ | $7.0^{\pm.2}$ | $.007^{\pm.001}$ | $7.3^{\pm.2}$ | $.008^{\pm.001}$ | $7.7^{\pm.3}$ | $.008^{\pm.001}$ | $7.5^{\pm.3}$ | $.005^{\pm.001}$ | $7.0^{\pm.5}$ | $.004^{\pm.002}$ |
| | | | | | | Sensitive feature not in the functional form of the model | | | | | | | | | |
| New Tasks | SCH | $7.7^{\pm.8}$ | $.088^{\pm.003}$ | $13.5^{\pm.3}$ | $.026^{\pm.001}$ | $12.9^{\pm.5}$ | $.023^{\pm.001}$ | $13.4^{\pm.4}$ | $.032^{\pm.002}$ | $9.9^{\pm.4}$ | $.018^{\pm.001}$ | $10.1^{\pm.5}$ | $.017^{\pm.002}$ | $10.2^{\pm.5}$ | $.017^{\pm.003}$ |
| | UNI | $9.1^{\pm.8}$ | $.175^{\pm.003}$ | $12.1^{\pm.7}$ | $.142^{\pm.007}$ | $12.9^{\pm.8}$ | $.160^{\pm.011}$ | $11.0^{\pm.6}$ | $.101^{\pm.003}$ | $11.7^{\pm.6}$ | $.136^{\pm.007}$ | $11.9^{\pm.6}$ | $.126^{\pm.009}$ | $11.6^{\pm.6}$ | $.115^{\pm.009}$ |
| | MOV | $7.9^{\pm.6}$ | $.128^{\pm.007}$ | $10.8^{\pm.4}$ | $.012^{\pm.001}$ | $11.4^{\pm.5}$ | $.018^{\pm.002}$ | $11.5^{\pm.5}$ | $.022^{\pm.003}$ | $10.3^{\pm.4}$ | $.012^{\pm.001}$ | $10.1^{\pm.5}$ | $.009^{\pm.001}$ | $9.8^{\pm.5}$ | $.008^{\pm.001}$ |
| | | | | | | Sensitive feature in the functional form of the model | | | | | | | | | |
| | SCH | $7.6^{\pm.5}$ | $.096^{\pm.005}$ | $12.1^{\pm.4}$ | $.032^{\pm.001}$ | $11.9^{\pm.4}$ | $.034^{\pm.002}$ | $11.5^{\pm.5}$ | $.025^{\pm.001}$ | $9.2^{\pm.4}$ | $.024^{\pm.001}$ | $9.0^{\pm.4}$ | $.018^{\pm.002}$ | $8.8^{\pm.5}$ | $.016^{\pm.002}$ |
| | UNI | $8.9^{\pm.6}$ | $.212^{\pm.001}$ | $11.9^{\pm.7}$ | $.241^{\pm.014}$ | $10.3^{\pm.4}$ | $.191^{\pm.009}$ | $11.1^{\pm.6}$ | $.221^{\pm.012}$ | $10.8^{\pm.4}$ | $.200^{\pm.011}$ | $10.8^{\pm.5}$ | $.162^{\pm.010}$ | $10.8^{\pm.4}$ | $.135^{\pm.010}$ |
| | MOV | $7.8^{\pm.7}$ | $.139^{\pm.011}$ | $9.9^{\pm.5}$ | $.022^{\pm.001}$ | $10.2^{\pm.7}$ | $.028^{\pm.003}$ | $10.1^{\pm.6}$ | $.024^{\pm.003}$ | $9.2^{\pm.4}$ | $.020^{\pm.001}$ | $9.3^{\pm.4}$ | $.018^{\pm.003}$ | $9.9^{\pm.5}$ | $.019^{\pm.002}$ |

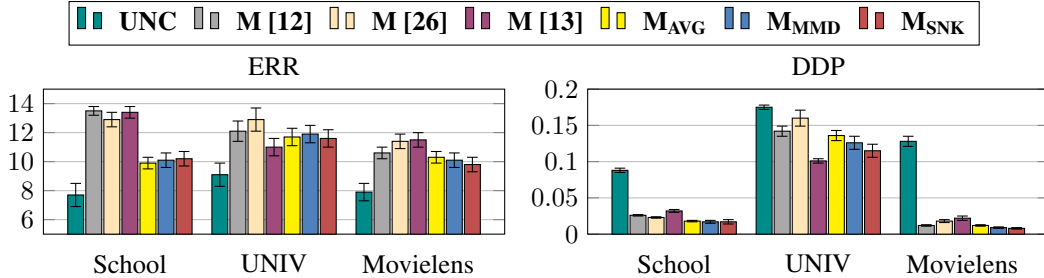

Figure 2: Graphical representation of the results in Table 2 for new tasks when the sensitive feature is not included in the functional form of the model.

**Discussion.** Our experimental results offer several interesting aspects and comparisons. Unsurprisingly, the lowest error (ERR) is reached using the unconstrained method (UNC), where we obtain very unfair models, i.e., with high DDP among all the datasets (UNC can be considered as gold standard for minimum error). Concerning the constrained methods, it is possible to note how – in general – learning a fair shared representation slightly increases the final error but brings a large decrease of the unfairness. In particular, we observe that this benefit is maintained also by tackling new and unseen (during the training of the shared representation) tasks. The same analysis of the results applies to both having and not having the sensitive feature in the functional form of the model. Comparing our methodology with other state-of-the-art techniques, we note how our proposals ($M_{AVG}$, $M_{MMD}$, and $M_{SNK}$), in all the settings, obtain better or comparable performance. In fact, our methods are able to maintain a larger accuracy and simultaneously a smaller fairness risk. In particular, $M_{MMD}$ and $M_{SNK}$ seem to produce better models than $M_{AVG}$, and all three perform better than state-of-the-art methods.

# 6 Conclusion

We have presented a method to learn a fair shared representation among different tasks in a MTL setting. Our method provides good generalization performance both in accuracy and fairness over novel and unseen tasks. We studied the learning ability of our method and we analyzed the performance over several experimental scenarios. The obtained results corroborate our theoretical findings and proved that our approach overcomes common benchmark algorithms and current state-of-the-art methods. Our next step will be to study (explicit) fair representation learning in the context of deep neural networks, with particular attention to the interpretability and transparency of the learned representation.

## Broader impact

Algorithmic fairness has a potential high social importance. The goal is to make safer the application of automatic agents as decision makers in our society. We think that learning a fair representation can be a practical way to pursue the goal of generating unbiased machine learning. A fair machine learning is needed in our society, especially after several discoveries of unfair biases in the current standard machine learning models. With less biased and more fair machine learning models, we can increase the trust of people in automatic agents – and we can also spread awareness of the possible issue of bias in machine learning models among colleagues in our research community. We have the possibility to enhance the benefits that using machine learning can provide to society and we need to avoid translating the negative human biases to the learned models.

We are aware that statistical measures of fairness (such as statistical parity or equal opportunity) cannot be considered as the unique definitions for bias. In fact, many others have been presented, exploring areas like – for example – causality. Indeed, we know that the choice of a definition of fairness for the task at hand has to be carefully understood by the user (i.e., a human) and not selected by an automatic agent. In this sense, it is well known that different definitions of fairness are even in contrast one each other. Consequently, enforcing one definition, we are simultaneously forcing other definitions to be violated. The choice of the right definition is fundamental but it is out of the scope of our proposal, and requires a careful human-in-the-loop approach.

## Acknowledgments and Disclosure of Funding

This work was supported by AWS Amazon Research Awards and SAP SE.

## Footnotes

[1]See for example the Caffe Model Zoo: `github.com/BVLC/caffe/wiki/Model-Zoo`

[2] Our method naturally extends to multiple sensitive variables but to ease the presentation we consider only the binary case in the paper.

[3] Depending on the application at hand, the representation may include also the sensitive feature in its functional form. In this case we just add two more components to $x$, representing the one-hot encoding of $s$ and proceed with our analysis as in the paper. However, for simplicity throughout we consider the case that $\mathcal{X} = \mathbb{R}^d$.

[4] In contrast the biased estimator is obtained by including the diagonal terms in the first two sums in (3) and renormalizing by $n^2$ and $m^2$, respectively.

[5]This assumption is made to simplify the presentation but is not a restriction to our analysis. In the general case our bound will be governed by the smallest sample size and have a similar flavour.

[6]The data and the research are related to the project DROP@UNIGE of the University of Genoa.

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
