[Supplementary Material]

## Appendix

In this appendix we first present some auxiliary results and then give the proofs of the results stated in the main body of the paper, which we restate here for the convenience of the reader.

The following lemma gives a bound on the Lipschitz constant of the MMD$^2$ estimator, that we use in the proof of Corollary 2.

**Lemma 4.** *The MMD estimator (3) has Lipschitz constant $8L_K/\sqrt{n}$ w.r.t. the Euclidean norm, where $L_K$ is the Lipschitz constant of the kernel function.*

**Proof.** We only consider one term in the computation of the Lipschitz constant, the other terms being conceptually identical. We have

$$\frac{1}{n(n-1)} \sum_{i \neq j} K(x_i, x_j) - \frac{1}{n(n-1)} \sum_{i \neq j} K(x_i', x_j') = \frac{1}{n(n-1)} \sum_{i \neq j} \left( K(x_i, x_j) - K(x_i', x_j') \right).$$

We add and subtract $K(x_i, x_j')$ inside the sum and rearrange the terms, so that

$$
\begin{aligned}
K(x_i, x_j) - K(x_i', x_j') &= K(x_i, x_j) - K(x_i, x_j') + K(x_i, x_j') - K(x_i', x_j') \\
&\leq L_K \|x_j - x_j'\| + L_K \|x_i - x_i'\|.
\end{aligned}
$$

Thus

$$
\begin{aligned}
\frac{1}{n(n-1)} \sum_{i \neq j} \left( K(x_i, x_j) - K(x_i', x_j') \right) &\leq \frac{L_K}{n(n-1)} \left( (n-1) \sum_{j=1}^{n} \|x_j - x_j'\| + (n-1) \sum_{i=1}^{n} \|x_i - x_i'\| \right) \\
&= \frac{2L_K}{n} \sum_{i=1}^{n} \|x_i - x_i'\| \leq \frac{2L_K}{\sqrt{n}} \|\mathbf{x} - \mathbf{x}'\|.
\end{aligned}
$$

We then repeat the argument for the other two terms in (3), contributing for $2L_K/\sqrt{n}$ and $4L_K/\sqrt{n}$, respectively.

∎

We note that the same result holds for the $V$-statistic estimator of MMD$^2$ mentioned below equation (3). As noted this estimator has an $O(1/n)$ bias, so it is less appealing in the context of Theorem 1 and Corollary 2.

**Proposition 5** (Rademacher bound for linear representations). *Let $\mathcal{X} = \{x \in \mathbb{R}^d : \|x\| \leq B\}$. Consider the class of representations $\mathcal{H} = \{h : \mathcal{X} \to \mathbb{R}^r, \ h(x) = \sigma(Wx) : \|W\|_F \leq 1\}$, where the activation function $\sigma : \mathbb{R} \to \mathbb{R}$ is Lipschitz with constant $L_\sigma$. Let $\mathcal{H}(\mathbf{X}) = \{h(x_{ti}) : t \in \{1, \dots, T\}, \ i \in \{1, \dots, n\}\}$. Then*

$$R(\mathcal{H}(\mathbf{X})) = \mathbb{E}_\epsilon \sup_{h \in \mathcal{H}} \sum_{t=1}^{T} \sum_{i=1}^{n} \sum_{k=1}^{r} \epsilon_{t,i,k} h_k(x_{t,i}) = L_\sigma B \sqrt{rnT}.$$

**Proof.** We have

$$
\begin{aligned}
R(\mathcal{H}(\mathbf{X})) &= \mathbb{E}_\epsilon \sup_{\|A\|_F \le 1} \sum_{t=1}^{T} \sum_{i=1}^{n} \sum_{k=1}^{r} \epsilon_{t,i,k} \sigma(\langle a_k, x_{t,i} \rangle) \\
&\le \mathbb{E}_\epsilon L_\sigma \sup_{\|A\|_F \le 1} \sum_{t=1}^{T} \sum_{i=1}^{n} \sum_{k=1}^{r} \epsilon_{t,i,k} \langle a_k, x_{t,i} \rangle \\
&= \mathbb{E}_\epsilon L_\sigma \sup_{\|A\|_F \le 1} \sum_{k=1}^{r} \langle a_k, \sum_{t=1}^{T} \sum_{i=1}^{n} \epsilon_{t,i,k} x_{t,i} \rangle \\
&\le \mathbb{E}_\epsilon L_\sigma \sqrt{ \sum_{k=1}^{r} \left\| \sum_{t=1}^{T} \sum_{i=1}^{n} \epsilon_{t,i,k} x_{t,i} \right\|^2 } \\
&\le L_\sigma \sqrt{ \mathbb{E}_\epsilon \sum_{k=1}^{r} \left\| \sum_{t=1}^{T} \sum_{i=1}^{n} \epsilon_{t,i,k} x_{t,i} \right\|^2 } \\
&= L_\sigma \sqrt{ \sum_{k=1}^{r} \sum_{t=1}^{T} \sum_{i=1}^{n} \|x_{t,i}\|^2 } \le L_\sigma B \sqrt{rTn}
\end{aligned}
\tag{12}
$$

where the first inequality follows by the standard contraction inequality, the second inequality by Cauchy-Schwarz's inequality, and the third inequality uses Jensen's inequality. ∎

**Theorem 1.** *Let $d$ be the unbiased $\mathrm{MMD}^2$ estimator (3). Let $(P_1, Q_1), \ldots, (P_T, Q_T)$ be independently sampled from $\rho$ and, for every $t \in \{1, \ldots, T\}$, let $\mathbf{x}_t \sim P_t^n$ and $\mathbf{z}_t \sim Q_t^n$. Then it holds with probability at least $1 - \delta$ in the draw of the multi-sample $(\mathbf{X}, \mathbf{Z}) = (\mathbf{x}_t, \mathbf{z}_t)_{t=1}^{T}$, that*

$$
\sup_{h \in \mathcal{H}} \left\{ \mathbb{E}_{(P,Q) \sim \rho} d(h_\sharp P, h_\sharp Q) - \frac{1}{T} \sum_{t=1}^{T} d(h(\mathbf{x}_t), h(\mathbf{z}_t)) \right\} \le C_1 \frac{L_d}{T} R(\mathcal{H}(\mathbf{X}, \mathbf{Z})) + C_2 \sqrt{\frac{\log 1/\delta}{T}} \tag{10}
$$

*where $L_d$ is the Lipschitz constant of the mapping $(\mathbf{x}, \mathbf{z}) \mapsto d(\mathbf{x}, \mathbf{z})$ w.r.t. the Euclidean norm and $C_1$ and $C_2$ are small positive numerical constants.*

**Proof.** We add and subtract the term

$$
\mathbb{E}_{(P,Q) \sim \rho} \mathbb{E}_{(\mathbf{x}, \mathbf{z}) \sim (P \times Q)^n} d(h(\mathbf{x}), h(\mathbf{z})) \tag{13}
$$

inside the supremum in the l.h.s. of (10) and decompose it as the sum of two terms, $\sup_h A_h + \sup_h B_h$, where

$$
A_h = \mathbb{E}_{(P,Q) \sim \rho} \mathbb{E}_{(\mathbf{x}, \mathbf{z}) \sim (P \times Q)^n} \left[ d(h_\# P, h_\# Q) - d(h(\mathbf{x}), h(\mathbf{z})) \right]
$$

and

$$
B_h = \mathbb{E}_{(P,Q) \sim \rho} \mathbb{E}_{(\mathbf{x}, \mathbf{z}) \sim (P \times Q)^n} d(h(\mathbf{x}), h(\mathbf{z})) - \frac{1}{T} \sum_{t=1}^{T} d(h(\mathbf{x}_t), h(\mathbf{z}_t)).
$$

For a fixed representation $h$, $A_h$ measures the average bias of the estimator over the draw of the distributions from the environment and their samples from the environment. For the $\mathrm{MMD}^2$ estimator (3) this term is zero, since the estimator is unbiased. Indeed

$$
\begin{aligned}
\mathbb{E}_{(\mathbf{x}, \mathbf{z}) \sim (P \times Q)^n} d(h(\mathbf{x}), h(\mathbf{z})) &= \mathbb{E}_{x, x' \sim P^2} K(x, x') + \mathbb{E}_{y, y' \sim Q^2} K(y, y') - 2 \mathbb{E}_{x \sim P, y \sim Q} K(x, y) \\
&= \mathbb{E}_{x, x' \sim P^2} \mathbb{E}_{y, y' \sim Q^2} \langle \Psi(x) - \Psi(y), \Psi(x') - \Psi(y') \rangle_{\mathbb{H}} \\
&= \langle \mathbb{E}_{x \sim P} \Psi(x) - \mathbb{E}_{y \sim Q} \Psi(y), \mathbb{E}_{x' \sim P} \Psi(x') - \mathbb{E}_{y' \sim Q} \Psi(y') \rangle_{\mathbb{H}} \\
&= \mathrm{MMD}^2(P, Q).
\end{aligned}
$$

The term $\sup_h B_h$ can be bounded with a uniform Rademacher bound, see e.g., [4]. Specifically, noting the term (13) can be interpreted as the expectation over the random variable $(\mathbf{x}, \mathbf{z}) \sim \hat{\rho}$, where the probability measure $\hat{\rho}$ models the draw of a random sample from the environment, we have that

$$\sup_{h \in \mathcal{H}} B_h \leq \frac{2}{T} \mathbb{E}_{\varepsilon} \sup_{h \in \mathcal{H}} \sum_{t=1}^{T} \epsilon_t d(h(\mathbf{x}_t), h(\mathbf{z}_t)) + \sqrt{\frac{\log 1/\delta}{2T}}$$

where $\epsilon_1, \ldots, \epsilon_T$ are i.i.d. Rademacher. The first term in the r.h.s. of the above inequality is the Rademacher complexity of the set

$$\Big\{ \big(d(h(\mathbf{x}_1), h(\mathbf{z}_1)), \ldots, d(h(\mathbf{x}_T), h(\mathbf{z}_T))\big) : h \in \mathcal{H} \Big\}.$$

Using the vector contraction inequality [27, Ineq. (1)] we can factor out the Lipschitz constant of the function $d$ and consider the Rademacher complexity of the set $\mathcal{H}(\mathbf{X}, \mathbf{Z})$. That is,

$$\mathbb{E}_{\varepsilon} \sup_{h \in \mathcal{H}} \sum_{t=1}^{T} \epsilon_t d(h(\mathbf{x}_t), h(\mathbf{z}_t)) \leq L_d R\big(\mathcal{H}(\mathbf{X}, \mathbf{Z})\big).$$

The result follows. ∎

**Corollary 2.** *Consider the class of representations* (7)*, assume that the activation function is Lipschitz with constant* $L_\sigma$ *and that the matrix* $W$ *of parameters has Frobenius norm bounded by* 1*. Assume further that* $\|x\| \leq B$ *a.s. over the environment* $\rho$*. Then under the same assumptions in Theorem 1 it holds that*

$$\mathbb{E}_{(P,Q) \sim \rho} d(h_\sharp P, h_\sharp Q) - \frac{1}{T} \sum_{t=1}^{T} d(h(\mathbf{x}_t), h(\mathbf{z}_t)) \leq C_1 L_\sigma B \sqrt{\frac{r}{T}} + C_2 \sqrt{\frac{\log 1/\delta}{T}}.$$

**Proof.** The proof follows by combing Theorem 1 with Lemma 4 and Proposition 5. ∎

**Lemma 3.** *Let* $\bar{h}$ *be the representation learned by method* (8) *and let with some abuse of notation* $\bar{P}, \bar{Q} \in \mathcal{P}(\mathbb{R}^r)$ *be the probability distributions associated to the two sensitive groups after the raw input is preprocessed with* $\bar{h}$*. Let the* MMD *distance at the representation level be measured w.r.t. kernel* $K : \mathbb{R}^r \times \mathbb{R}^r \to \mathbb{R}$ *and the* MMD *on the output be measured w.r.t. kernel* $G : \mathbb{R} \times \mathbb{R} \to \mathbb{R}$*. Then, for every* $v \in \mathbb{R}^r$ *we have*

$$\mathrm{MMD}_G(v_\# \bar{P}, v_\# \bar{Q}) \leq \sup_{\|g\|_G \leq 1} \|g(\langle v, \cdot \rangle)\|_K \mathrm{MMD}_K(\bar{P}, \bar{Q}).$$

**Proof.** We have

$$\mathrm{MMD}_G(v_\# \bar{P}, v_\# \bar{Q}) = \left\| \int G(\xi, \cdot) \, d(v_\# \bar{P})(\xi) - \int G(\xi, \cdot) \, d(v_\# \bar{Q})(\xi) \right\|_G \tag{14}$$

$$= \left\| \int G(\langle v, z \rangle, \cdot) \, d\bar{P}(z) - \int G(\langle v, z \rangle, \cdot) \, d\bar{Q}(z) \right\|_G. \tag{15}$$

Hence, using the formula $\|f\|_G = \sup_{\|g\|_G \leq 1} \langle g, f \rangle_G$,

$$\mathrm{MMD}_G(v_\# \bar{P}, v_\# \bar{Q}) = \sup_{\|g\|_G \leq 1} \langle g, \int G(\langle v, z \rangle, \cdot) \, d\bar{P}(z) - \int G(\langle v, z \rangle, \cdot) \, d\bar{Q}(z) \rangle_G \tag{16}$$

$$= \sup_{\|g\|_G \leq 1} \int g(\langle v, z \rangle) \, d(\bar{P} - \bar{Q})(z) \tag{17}$$

$$\leq \sup_{\|g\|_G \leq 1} \|g \circ v\|_K \mathrm{MMD}_K(\bar{P}, \bar{Q}). \tag{18}$$

The result follows. ∎

A full analysis for Sinkhorn divergence rather than MMD distance is left for future work. Here we provide preliminary tools that we plan to use to develop the analogous of the results above in the case of Optimal Transport distances. In particular, the proposition below appears as the counterpart of Lemma 3. In this case, the dependency on the action of the pushforward seems to be neater and more explicit than in MMD case.

**Proposition 6.** *Let $\mathcal{X} \subset \mathbb{R}^d$ and $P, Q \in \mathcal{P}(\mathcal{X})$. Let $T : \mathcal{X} \to \mathcal{X}$ a Lipschitz map with Lipschitz constant L. Then*

$$\mathsf{OT}(T_\# P, T_\# Q) \leq \max(L^2, 1)\mathsf{OT}(P, Q). \tag{19}$$

To prove the proposition above we need the following lemma. We introduce some notation first, writing explicitly the dependence on the cost function:

$$\mathsf{OT}_{\varepsilon, \|\cdot\|^2}(P, Q) = \min_{\pi \in \Pi(P,Q)} \int_{\mathcal{X}^2} \|x - y\|^2 \, d\pi(x,y) + \varepsilon \mathsf{KL}(\pi | P \otimes Q)$$

and

$$\mathsf{OT}_{\varepsilon, \|T(\cdot)\|^2}(P, Q) = \min_{\pi \in \Pi(P,Q)} \int_{\mathcal{X}^2} \|T(x) - T(y)\|^2 \, d\pi(x,y) + \varepsilon \mathsf{KL}(\pi | P \otimes Q). \tag{20}$$

Also, for a general cost function c, recall that $\mathsf{OT}_{\varepsilon, \mathsf{c}}$ has a dual formulation which reads as [15]

$$\mathsf{OT}_{\varepsilon, \mathsf{c}}(P, Q) = \sup_{f,g \in \mathcal{C}(\mathcal{X})} \int f \, dP + \int g \, dQ - \varepsilon \int e^{\frac{f(x)+g(y)-\mathsf{c}(x,y)}{\varepsilon}} \, dP(x) \, dQ(y)$$

where $\mathcal{C}(\mathcal{X})$ is the set of continuous functions on $\mathcal{X}$.

**Lemma 7.** *Let $P, Q \in \mathcal{P}(\mathcal{X})$ and $T : \mathcal{X} \to \mathcal{X}$ be a continuous map. Then,*

$$\mathsf{OT}_{\varepsilon, \|\cdot\|^2}(T_\# P, T_\# Q) = \mathsf{OT}_{\varepsilon, \|T(\cdot)\|^2}(P, Q). \tag{21}$$

**Proof.** Let $F(P, Q, f, g, \|\cdot\|^2)$ be defined as

$$F(P, Q, f, g, \|\cdot\|^2) = \int_{\mathcal{X}} f(x) \, dP(x) + \int_{\mathcal{X}} g(y) \, dQ(y) - \varepsilon \int e^{\frac{f(x)+g(y)-\|x-y\|^2}{\varepsilon}} \, dP(x)dQ(y).$$

By the dual definition of $\mathsf{OT}_{\varepsilon, \|\cdot\|^2}$ and the property of pushforward measures (see [2, Sec 5.2], we have

$$\mathsf{OT}_{\varepsilon, \|\cdot\|^2}(T_\# P, T_\# Q) = \sup_{(f,g) \in \mathcal{C}(\mathcal{X}) \times \mathcal{C}(\mathcal{X})} F(T_\# Q, T_\# Q, f, g, \|\cdot\|^2) \tag{22}$$

$$= \sup_{(f,g) \in \mathcal{C}(\mathcal{X}) \times \mathcal{C}(\mathcal{X})} F(P, Q, f \circ T, g \circ T, \|T(\cdot)\|^2) \tag{23}$$

$$= \sup_{(\tilde{f},\tilde{g}) \in (\mathcal{C}(\mathcal{X}) \circ T) \times (\mathcal{C}(\mathcal{X}) \circ T)} F(P, Q, \tilde{f}, \tilde{g}, \|T(\cdot)\|^2) \tag{24}$$

where $\mathcal{C}(\mathcal{X}) \circ T := \{u \circ T : u \in \mathcal{C}(\mathcal{X})\}$. Now, consider

$$\mathsf{OT}_{\varepsilon, \|\cdot\|^2}(P, Q) = \sup_{(\tilde{f},\tilde{g}) \in \mathcal{C}(\mathcal{X}) \times \mathcal{C}(\mathcal{X})} \int_{\mathcal{X}} \tilde{f}(x) \, dP(x) + \int_{\mathcal{X}} \tilde{g}(y) \, dQ(y) + \tag{25}$$

$$-\varepsilon \int e^{\frac{f(x)+g(y)-\|T(x)-T(y)\|^2}{\varepsilon}} \, dP(x)dQ(y).$$

We note that the optimal potentials $\tilde{f}, \tilde{g}$ of $\mathsf{OT}_{\varepsilon, \|T(\cdot)\|^2}$ have the form [15]

$$\tilde{f}(x) = -\log \int_{\mathcal{X}} e^{\tilde{g}(y) - \|T(x)-T(y)\|^2} \, dQ(y).$$

We note that $\tilde{f}$ and $\tilde{g}$ are functions of the form $u \circ T$ and $v \circ T$. Hence the supremum in (25) can be restricted to be on the set $\mathcal{C}(\mathcal{X}) \circ T$. Thus, the quantity in (24) equals $\mathsf{OT}_{\varepsilon, \|T(\cdot)\|^2}$, showing the desired result. ∎

We now prove Proposition 6.

**Proof of Proposition 6.** Thanks to Lemma 7, we have that $\mathsf{OT}_{\varepsilon,\|\cdot\|^2}(T_\# P, T_\# Q) = \mathsf{OT}_{\varepsilon,\|T(\cdot)\|^2}(P,Q)$. Using this fact together with the definition of $\mathsf{OT}_{\varepsilon,\|T(\cdot)\|^2}$ recalled in (20) and the Lipschitz property of $T$, we have

$$
\begin{aligned}
\mathsf{OT}_{\varepsilon,\|\cdot\|^2}(T_\# P, T_\# Q) &= \mathsf{OT}_{\varepsilon,\|T(\cdot)\|^2}(P,Q) \\
&= \min_{\pi \in \Pi(P,Q)} \int_{\mathcal{X}^2} \|T(x) - T(y)\|^2 \, d\pi(x,y) + \varepsilon \mathrm{KL}(\pi | P \otimes Q) \\
&\leq \min_{\pi \in \Pi(P,Q)} \int_{\mathcal{X}^2} L^2 \|x - y\|^2 \, d\pi(x,y) + \varepsilon \mathrm{KL}(\pi | P \otimes Q) \\
&\leq \max(L^2, 1) \min_{\pi \in \Pi(P,Q)} \int_{\mathcal{X}^2} \|x - y\|^2 \, d\pi(x,y) + \varepsilon \mathrm{KL}(\pi | P \otimes Q) \\
&= \max(L^2, 1) \mathsf{OT}_{\varepsilon,\|\cdot\|^2}(P,Q).
\end{aligned}
$$

∎