[Reviews · NeurIPS 2020]

Review 1

Summary and Contributions: EDIT: Thank you for the rebuttal. I appreciate you pointing out the "sensitive attribute included" portion of the experiments; it's good that you included that, and would be good to have an accompanying figure in the appendix (or main body, space permitting). My main sticking point is that, although the paper claims to learn representations which can transfer to new tasks, there are no theoretical results which address the model performance on new tasks. This seems like a large hole, and without those results the paper is definitely still marginal in my opinion; I worry that readers will not get much out of it. However, it is well put together and the contributions provided are real - so averaging that all out, my score remains at 6. -------------------------------------------------- - This paper gives results on fair representation learning, considering a multi-task learning setup, including a generalization bound to new input tasks on the fairness of representations in a binary sensitive attribute context.

Strengths: -- This is a useful idea: given some new task, how fair will be expect our learned representation function to be? Very relevant to the fair representation learning NeurIPS community - Theoretical results seems sound and answer a reasonable question about how fair representations might be given new distributions on the inputs and sensitive attributes - Experiments demonstrate that in the proposed setting, the method seems to be useful and better than some baselines

Weaknesses: -- It’s not clear to me that the paper’s theoretical results on transfer learning are that interesting, since they do not discuss shifts in P(Y | X, S), which will not affect fairness but will affect accuracy. Rather, I would characterize the results as on generalization of fair representations under covariate shift of (X, S) - Experiments are not particularly broad, and not necessarily the best comparison to baselines (weak adversary and no sensitive attribute included). But I’m not sure they are that important in this paper anyways

Correctness: -- Yes this paper seems correct.

Clarity: -- Mostly clear

Relation to Prior Work: -- Yes

Reproducibility: No

Additional Feedback: -- Would be nice to see the code - Should be a clarification here: since the fairness definition considered is label independent, when we talk about transferring performance to new tasks we are not concerned with a shift in the label distribution (marginal or conditional). This is a pretty important point which is not mentioned at all - Related work: would be nice to see more differentiation between different approaches in the current literature (for instance [4, 11, 23, 24, 27, 28, 37, 19, 40] are not all identical papers and present different approaches) - L140: clarify that the non-linearity is from dimension r to dimension r. As of now, it looks like h has a scalar output (which I know is not true) - L159: clarify what “Gap” is - L163: “approximately satisfied” – should clarify what this means - L199: Proposition 5 should be an inequality rather than an equality right? - In general should at least give a note on accuracy – would we expect accuracy to degrade at all? - L221: The (M [24]) notation is very confusing, not sure why you’re using this - L238: The scheme for choosing the model is interesting, although I prefer to see the full tradeoff between fairness and accuracy - Experiments: I’m not sure that the comparison to [11] and [24] is very fair, since those methods usually expect the sensitive attribute to be part of the data, and also require the usage of a powerful adversary to be useful - L253: Not sure why the University will be anonymized? - Conclusion: it feels like we didn’t fully explore transfer learning here, since different distributions P(Y | X, S) are extremely important in transfer learning, and shifts in that distribution don’t figure in to the theoretical results at all. - It was disappointing to me there are no results on fairness/accuracy tradeoff in the generalization setting


Review 2

Summary and Contributions: Post author response: Thanks for the clarifying remarks. I agree that DP is a reasonable starting point for such work. I particularly appreciate the explanation of the potential relevance of the setup, which I'd love to see discussed in a later version of the paper. While the other reviewers have raised some interesting points (in particular about the distinction of shifts in (X, Y) vs P(Y | X, S) which are worthy of a brief discussion as well), I still believe this submission would be of value to the NeurIPS community and will stick to my original score. --------------- This submission develops theory and methodology for learning demographically fair representations in a multitask setting, such that the learned representations transfer to new tasks from a fixed distribution. It establishes bounds on demographic parity on new tasks and demonstrates the effectiveness of their MMD and sinkhorn based algorithms on various real-world datasets.

Strengths: Both the setup and theoretical claims are stated clearly and as far as I can tell correct. The empirical evaluation is thought through well, includes relevant comparisons to existing work and demonstrates the effectiveness of the algorithms (some comments about empirical evaluation in "Additional feedback" section.) I believe the contributions of this submission are novel and in general relevant to the NeurIPS community (see "Weaknesses" section for some potential caveats in terms of the relevance for fairness).

Weaknesses: * In my opinion, the biggest weakness is that it is not entirely clear how the contributions advance our understanding and practices of fair machine learning in applications. First, the submission focuses entirely on demographic parity, which is rarely ever considered as a standalone criterion to ensure fairness in practice. Moreover, as a community we are currently grappling with how, when, and where fairness enforcing algorithms may or may not make sense at all, swiftly moving away from context-blind statistical group fairness measures. The present submission further specializes these criteria for a fictitious multitask setting over the same base space. Against this backdrop, it is hard for me to envision scenarios in which the devised methods could have real-world impact. It would be great, if the authors could share their thoughts on whether and when these techniques may be useful and deployed with good conscious. * My second concern is related to the first one: I did not fully understand how the multitask setting is realized in the empirical experiments on static supervised datasets? What is the distribution $\rho$? How many tasks are there for each dataset and how are they defined? However, on a theoretical and methodological level I still believe that the shown contributions are significant and interesting enough to warrant publication at NeurIPS.

Correctness: I did not check all proofs in detail, but from the well-written proof sketches and glancing over the supplementary material, I believe that all claims in the submission are correct. The methods for the empirical evaluation seem to be valid, in particular the authors are careful about model selection and provide some error quantification across three datasets, comparing to the most relevant baselines from existing work. However, I would have wished for more details explaining the precise setup of the empirical evaluation for each dataset, including a description of $\rho$ and the different tasks used as well as training details (network sizes, optimizers, etc.). (Due to space restrictions, this can be moved to the appendix in my opinion.) Ideally, I would like to see an implementation to ensure reproducibility and a fair comparison to baselines. As it stands, there is not sufficient detail to reproduce the empirical results.

Clarity: The paper is very well written and structured. The only comment I have is that the meta-distribution $\rho$ over tasks is only mentioned in l.152. This should be moved to the first mention of "generalization to new tasks" (e.g. l.104 or even earlier). I was confused about this statement at first since surely some assumption on the similarity or distribution of tasks is required. I especially enjoyed the concise, yet rigorous, introduction of the setup and required assumptions as well as the recap of MMD and Sinkhorn divergence.

Relation to Prior Work: Relevant existing work is acknowledged appropriately to the best of my knowledge. Instead of the generic long list of references in l.65 (and what these works try to achieve in general), I would have liked to see a brief discussion of whether and how the presented methodology deviates from existing work. It seems to me that beyond having introduced multiple tasks, the methods of minimizing a similarity measure such as MMD, KL, Sinkhorn, etc. in a gradient-based fashion is quite similar to existing work?

Reproducibility: No

Additional Feedback: * I liked how the empirical estimators for MMD are described in detail. How is the Sinkhorn divergence in eq. (5) estimated from finite samples? * The restriction to 1-hidden layer networks as well as linear functions g (or general functions of linear projections) starting in l.128 came as a surprise to me, as it seems at least some of the following claims hold more generally? Can the authors comment on whether that was merely a choice for the empirical evaluation and to what extent the theoretical considerations depend on these assumptions? The linear projections seem to be necessary for some parts (e.g., bound outcome fairness from representation fairness), but could I just drop-in an k-layer NN and obtain similar results (theoretically and empirically)? * l.188: What are the major complications in extending Theorem 1 to Sinkhorn divergence? Have the authors tried it? From the abstract and introduction it seems that theoretical guarantees are provided for both MMD and Sinkhorn, this should be made more explicit. * typo: eq. (2) both expectations are over Q * typo: l.159 in termS of... * typo: l.188: in order to extenD Theorem...


Review 3

Summary and Contributions: In this paper, the authors proposed a novel method to learn a fair shared representation among different tasks in a multi-task learning setting. The final objective function is composed of two terms: the loss term is represented by square or logistic loss while the fairness part is evaluated via Sinkhorn divergence and maximum mean discrepancy (MMD). The experimental results on three datasets could verify its effectiveness.

Strengths: 1. The paper focus on building fair and accurate models across different tasks, which is a very important problem in machine learning community. 2. Applying multi-task learning framework to leverage task similarities for fair representation learning. 3. The experimental results on three public datasets shows the proposed method is superior to its competitors.

Weaknesses: 1. The manuscript is difficult to read and the problem was not clearly stated, which makes it hard to evaluate the novelty and applicability of the method beyond the specific data sets evaluated here. Specifically, what is the main contribution of this study? Is it the first study that learn fair representation in the scenario of multi-task learning? 2. It is good to see that the authors have provided the learning bound for the objective function. However, little comment is given on the provided theorems to connect them to the problem context or the observed results, so the significance of the Theorem 1 is not well justified. 3. The contribution and innovation of the method are insufficient for the NeurIPS conference. Many of the methods involved are existing such as MMD, SNK. 4. There are many formula mistakes throughout the paper. For instance, Eq. (2) should be \|Ex~P \fi(X) − Ex~Q \fi(X)\|^2, the author should double-check them before submission. 5. Also, several symbol definitions are missed. For instance, what does the operator d(.) stand for in Eq.(6)?

Correctness: There are some mistakes in the formulas. The empirical methodology seems to be correct.

Clarity: The paper is difficult to follow and the main motivation was not clearly stated.

Relation to Prior Work: The difference between the proposed method and existing algorithm has not been clearly clarified.

Reproducibility: Yes

Additional Feedback: N/A.

[Author Response · NeurIPS 2020]

We thank the reviewers for their comments and for acknowledging that we address a relevant problem for the NeurIPS
community [R1,R2,R3], that our experiments show the utility / effectiveness of the proposed method [R1,R2,R3] and
that the setup and theoretical claims seem sound [R1,R2].
**R1** **Comparison to [11,24] not very fair:** Please note that the claimed missing result "sensitive attribute included"
***is in the paper*** (specifically see the columns in Tabs. 1 & 2 named "Sensitive feature in the functional form of
the model"); See also ll. 233-234. **They do not discuss shifts in $P(Y|X,S)$, which will not affect fairness but will**
**affect accuracy:** The reviewer raises a very interesting point. Note that our method (cf. Eq. (6)) looks for a shared low
complexity representation between the tasks. Shifts in $P(Y|X,S)$ are permitted, provided there is a common predictive
representation for the task outputs. Searching for such representation is indeed one of the main ideas behind transfer
learning (see e.g. Caruana 1997; Argyriou et al. 2008, etc.) and it is aided by the first term in Eq. (6), measuring the
average empirical risk on the training tasks. However, differently from the above works, our method also crucially
involves the second term, which encourages representations that approximately satisfies DP on average over the training
tasks. Our fairness violation bound (Thm 1) involves only the marginal distributions (of each sensitive group within the
task) since we care to measure this at the representation level, but see ll. 97-103 and Lemma 3 for how this affects DP at
the output level. In this sense our method learn a shared and transferable representation, one based on which accurate
and fair models can be learned on tasks sampled from the environment $\rho$. **Would we expect accuracy to degrade at**
**all?** Yes, accuracy decreases due to the fairness constraint. This property is common to any algorithmic fairness method.
**Other suggestions:** These will be addressed in the revision. In particular we'll: make our code publicly available upon
acceptance, add at l. 140 that $\sigma$ is applied component-wise, say at l. 159 that the function "Gap" is quantified by the
r.h.s. of Ineq. (10), clarify at l. 163 that "approximately" means that the marginal distributions of the two sensitive
groups within the task are closed according to some suitable measure (e.g. see that at l. 242), improve the "M [24]"
notation (it stays for "method [24]" but we could just write "[24]"). Finally concerning **fairness/accuracy tradeoff in**
**the generalization setting:** standard bounds from the learning-to-learn literature (e.g. [3]) can be readily used to bound
the risk (or accuracy) of the representation $h$ found by our method on future tasks by the minimal multitask empirical
risk (over the specifications $g_t$). Now since our method in Eq. (6) minimizes a tradeoff between the multitask empirical
risk and fairness violation, the larger $\gamma$ the larger the former term, so risk on future tasks will reflect this tradeoff too.
**R2** **Submission focuses entirely on dem. parity:** We agree DP is not the ultimate fairness notion. Still it is frequently
studied in the literature and our study is a valuable starting point for fair representation learning within the multitask
setting. **May techniques be deployed with good conscious?** Yes, our theoretical and experimental results give an
indication that the method could be valuable in practice and safely deployed. Of course more experiments would be
needed to assess its robustness on on real-world problems. **Fictitious multitask setting:** it is true the paper is more on
theory and methodology which is not always close to practice, but imagine the following real-life scenario: each task is
associated with an hospital in country X and the task is to predict whether a patient who visits emergency should be
hospitalized. The environment (meta-distribution) $\rho$ may be the uniform distribution (or weight larger hospitals more).
The sensitive attribute may be race and other non-sensitive variables may measure cough frequency and body temperature.
Our main result, Thm 1 (in conjunction with Cor 2) then says that if we use our method to learn a predictive representation
and observe it to be fair according to DP on the random training task datasets, then it will also be fair according to DP
on average on all possible hospitals at the population level (i.e. on average over random patients visiting the hospital),
which is a very appealing property. **Description of $\rho$:** Yes, we'll mentioning also at l. 104. **How is the multitask**
**setting realized in the empirical experiments?** Please see ll. 227-34: we test either on different data for the same
tasks using during training or on a new task in leave-one-task out setting. **How is the Sinkhorn divergence in eq. (5)**
**estimated from finite samples?** Consider $\hat{P}$ and $\hat{Q}$ the same as in ll. 116-117. Denote by $\mathsf{p} = (\mathsf{p}_1, \ldots, \mathsf{p}_n) := 1/n\mathbf{1}_n$
and $\mathsf{q} = (\mathsf{q}_1, \ldots, \mathsf{q}_m) := 1/m\mathbf{1}_m$, with $\mathbf{1}_k$ the vector with $k$ entries equal to one ($\mathsf{p}$ and $\mathsf{q}$ denote the weights of
the empirical distributions $\hat{P}$ and $\hat{Q}$). Then, $\mathsf{OT}_\varepsilon(\hat{P}, \hat{Q}) = \min_{T \in \Pi(\mathsf{p},\mathsf{q})} \langle T, C \rangle + \varepsilon \sum_{i,j=1}^{n,m} \log(T_{ij}/\mathsf{p}_i\mathsf{q}_j)T_{ij}$, where
$\Pi(\mathsf{p},\mathsf{q}) := \{T \in \mathbb{R}_+^{n \times m} \mid T\mathbf{1}_m = \mathsf{p}, T\mathbf{1}_n = \mathsf{q}\}$ and $C_{ij} = \|x_i - z_j\|^2$; see ref. [30] for more explanations. **Restriction**
**to 1-hidden layer nets:** Our method and Thm 1 apply to general classes of representation functions $h$ of suitably
bounded complexity. For simplicity we illustrated them on 1-hidden layer networks, both theoretically (Thm 1 + Cor 2)
and empirically. However, bounds in [Chain Rule for the Expected Suprema of Gaussian Processes, ALT 2014] could
be used in place of Cor 2 for multi-layer representations. **Extending Thm 1 to Sinkhorn divergence (l. 188):** There
are two main obstacles at this stage: first, the term $A_h$ at l. 425 would not be zero, because the estimators is biased.
Second, the Lipschitz behaviour needed to factor out the constant (see l. 436) is not clear in the case of Sinkhorn.
**R3** **1. Main contribution:** Please see l. 46. To the best of our knowledge this is the first paper using multitask learning
for fair representation. **2. Significance of Thm 1:** This result together with a bound on the Rademacher average of the
representation class (e.g. Cor. 2 in the case of linear representations) gives a justification for our method – see also the
reply to R2 concerning the point *fictitious multitask setting*. **3. Contribution insufficient:** We disagree: MMD and
SNK have been used only recently for algorithmic fairness. The proposed method is novel, empirically competitive and
theoretically grounded (see point 2 above). **4,5. Formula mistakes and missing definitions:** We'll carefully check
our formulas. Thanks for Eq. (2), but note $d(\cdot)$ is already defined at l. 109 of the paper.

[Meta-Review · NeurIPS 2020]

The paper focuses on finding fair (according to demographic parity) representations in multitask settings, which is indeed an interesting problem of the fair ML community. The paper is well written and the contributions are signifiant. That said, as pointed out by the reviewers in the (post-rebuttal updated) feedback, the paper could significantly increase its impact and contributions with a more thorough experimental evaluation, and additional theoretical results addressing the model performance on new tasks (although such extension could be also studied in future work). I encourage the authors to incorporate the reviewers' feedback in the revised version of the paper, which should especially clarify the distinction of shifts in (X, Y) vs P(Y | X, S) (see comments by R1 and R2).